# Numerical Analysis on Heat Characteristics of the Ventilation Basket for Fresh Tea Leaves

**DOI:** 10.3390/foods11152178

**Published:** 2022-07-22

**Authors:** Zhixiong Zeng, Yihong Jiang, Chengying Ma, Jin Chen, Xiaodan Zhang, Jicheng Lin, Yanhua Liu, Jiaming Guo

**Affiliations:** 1College of Engineering, South China Agricultural University, Guangzhou 510642, China; zhixzeng@scau.edu.cn (Z.Z.); piom000@163.com (Y.J.); jillchan@126.com (J.C.); 19303032472@163.com (X.Z.); linjicheng@stu.scau.edu.cn (J.L.); cynthialyh@126.com (Y.L.); 2Maoming Branch, Guangdong Laboratory for Lingnan Modern Agriculture, Guangzhou 525000, China; 3Tea Research Institute, Guangdong Academy of Agricultural Sciences, Guangzhou 510640, China; machengying@tea.gdaas.cn

**Keywords:** numerical simulation, structure optimization, temperature characteristics, fresh keeping

## Abstract

Plastic baskets are commonly used as containers for fresh tea leaves during storage and transport after harvest. Nevertheless, there are significant challenges in controlling the core temperature of the basket since fresh tea leaves still maintain a certain degree of respiration after being harvested, with extremely high temperatures being the major factor for the color change of fresh tea leaves. A numerical model was developed to improve the temperature control of the plastic basket, by which the influence of different structural parameters on the core temperature in the plastic baskets with fresh tea leaves was analyzed. The accuracy of the model in predicting airflow and temperature distributions was validated against experimental data. The maximum RMSE was 1.158 °C and the maximum MRE was 5.410% between the simulated and test temperature value. The maximum deviation between the simulated velocity and test velocity was 0.11 m/s, the maximum RE was 29.05% and the maximum SD was 0.024. The results show that a plastic basket with a ventilation duct efficiently decreased the temperature of the fresh tea leaves and significantly affected the heat transfer between the fresh tea leaves and the ambient air compared to the plastic basket without a ventilation duct. Furthermore, the effect on the heat transfer was further expanded by the use of a plastic basket with a ventilation duct when the plastic baskets were stacked. The maximum temperature differences were 0.52 and 0.40 according to the stacked and single-layer products, respectively. The ambient temperature and the bulk density of the fresh tea leaves have a significant influence on the core temperature.

## 1. Introduction

As one of the most popular beverages in the world, tea is sought after by people for its unique flavor and potential benefits [1,2,3]. Fresh tea contains numerous endogenous metabolites that are strongly associated with its quality, which are subject to post-harvest storage, processing technology, cultivation environment and growing season. Excessive aging or deterioration is among the most crucial factors affecting endogenous metabolites in tea after the leaves are harvested [4], implying that the suitable treatment of harvested fresh tea leaves has great effects on tea quality. Ambient temperature is an important factor in delaying the metabolic process in fresh tea leaves after harvesting. Various chemical reactions occur in the leaves [5,6] such as oxidation and hydrolysis. Sugars are decomposed with respiration, releasing a large amount of heat and leading to an increase in leaf temperature [7]. A number of tea polyphenols are hydrolyzed or oxidized due to the increase in fresh tea leaves’ temperature, which cause the reddening of the leaves [8], and the increase in respiration causes temperature fluctuations that affect the transpiration of fresh tea leaves and accelerates the loss of moisture content [9,10,11]. Therefore, it is necessary to control the ambient temperature to avoid extreme temperatures and to slow down the metabolic process in fresh tea leaves.

At present, a large amount of field heat-induced deterioration of fresh tea leaves occurs because transport baskets are overfilled. The bulk density of fresh tea leaves in a basket is considered to be the main factor for leaf temperature variation, and the deterioration of fresh tea leaves is accelerated due to field heat as well as mechanical damage. Low ambient temperature can reduce the oxidative decomposition of compounds and produce greater floral, sweet, and fruity characters, which show that a low temperature has a positive effect on maintaining the quality of tea [10,12]. Tulio et al. [13] measured the effects of different ambient temperatures on fresh tea leaves and found that the tea leaves were sensitive to chilling injury, and the best storage temperature was above 8 °C but below 15 °C. Therefore, in order to reduce the field heat and the inhibit compound decomposition and the physiological activity of leaves, it is necessary to prevent the accumulation of respiration heat from fresh tea leaves during storage or transportation to maintain the quality of fresh tea leaves.

Transport containers are used for the temporary storage and transportation of gardening products after harvesting. At present, plastic baskets and corrugated cardboard boxes [14,15,16] are often used to load fresh tea leaves. Corrugated cardboard boxes are often used for transportation after harvesting, but they cause resource waste and are less sustainable because of their one-time use purpose [17]. Although a plastic basket can load more fresh tea leaves, it squeezes the leaves and has poor air permeability in its core. The heat from respiration cannot be dissipated in time, and the chemical reaction in the leaves will be further strengthened under environments with high temperatures, which often leads to the reddening of fresh tea leaves in the center which is unsuitable for long-distance transportation and long–term storage [13]. In recent years, a number of researchers have conducted in-depth research and analysis on ventilated containers [18,19,20]. The optimized designs of well–ventilated transportation baskets were performed to obtain better cooling efficiency and ventilation uniformity [21,22], which is helpful for delaying product quality degradation and regulating product temperature [23]. Therefore, the development of a new type of tea transport basket is of great significance for the maintenance of fresh tea quality.

Experimentation is an effective method to evaluate the heat characteristics of a ventilation basket for fresh tea leaves. Nevertheless, experiments often cost significant amounts of time and labor, and the effect of sensor precision on the results is highly significant. In recent years, computational fluid dynamics (CFD) has been widely used in cold chain studies [24,25,26]. Researchers approbate its powerful visualization function and accuracy in numerical simulations [27], and the combination of experimental results and numerical simulations can help more comprehensively analyze the airflow information and temperature distribution. Gruyters et al. [28] compared corrugated cardboard boxes and plastic crates to evaluate the overall cooling performance of the two boxes by CFD methods, and the results show that plastic boxes have much better development and research prospects.

In this study, a new type of transportation ventilation basket for fresh tea leaves was developed that performs better under heat characteristics. The effects of the transportation basket with different parameters and the environmental regulation of temperature variations on fresh tea leaves were investigated. Furthermore, this study also examined the heat characteristics and airflow distribution between the new and traditional transportation ventilation basket when the product was stacked. The results can help maintain the quality of fresh tea leaves after harvest before being processed.

## 2. Materials and Methods

### 2.1. Materials

A fresh-keeping container (1.9 length × 1.1 width × 1.5 m high) was adopted to simulate the ambient environment during storage and transportation [27]. The container was divided into two parts: a fresh-keeping area and a refrigeration area, as shown in Figure 1. When the fan is running, air from the fresh-keeping area reaches the refrigeration area through the air duct and is cooled by the refrigerating system.

A new transportation basket (0.6 length × 0.425 width × 0.35 m high) with a ventilation duct was designed and investigated for its performance on the fresh-keeping of fresh tea leaves. The effects of the ambient temperature, structural parameters of the ventilation duct and bulk density on the temperature performance of the basket were considered. The structural parameters of the basket are shown in Figure 2.

The temperature variations of the fresh tea leaves and the air flow in the container were obtained and analyzed by numerical methods using ANSYS 18.2.

### 2.2. Experimental Test

#### 2.2.1. Measurement of the Product’s Resistance

The product studied in this paper was a three-dimensional porous medium. The ventilation resistance effects of fresh tea leaves can be considered in such a model by determining the viscous resistance coefficient and the inertia resistance coefficient [29], which can be measured by tests.

The two resistance parameters were determined by the air pressure drop after the air passed through the product at different velocities, as measured by the ventilation resistance device (Figure 3), which included a frequency conversion fan, the product, a differential pressure gauge (intelligent digital micro-pressure meter EY-200A, range: −100 Pa–100 Pa, accuracy: 0.1 Pa; Shanghai Yiou Instrument Equipment Co., Ltd.) and an impeller anemometer (TESTO410I, range: 0.4–30 m/s, accuracy: ± (0.2 m/s +2% measurement value); Testo, Germany). The velocity of the fan was controlled by a variable frequency switch, with which six values of air velocity were set. The drop in air pressure after passing through the product at different velocities was measured by monitoring the pressure difference between the front and back holes, repeated 3 times.

#### 2.2.2. Experimental Verification

A test was conducted in the fresh-keeping container (at a 1:1 scale to the container) to evaluate the accuracy of the simulation models. The structure of the container and product stacking are shown in Figure 1. The temperature of the fresh tea leaves in the basket was measured using 4 temperature sensors (Testo174H, measuring range: −20~70 °C, accuracy: ±0.5 °C; Testo, Germany), and their arrangement is shown in Figure 4. The velocities in the ventilation duct of the basket were measured by 6 hot wire anemometers (Testo405i, measuring range: 0–30 m/s, accuracy: ± (0.2 m/s +2% of the measurement value); Testo, Germany), and their arrangement is shown in Figure 5. The velocities of the perforated clapboard were measured using an impeller anemometer (Testo410i, measuring range: 0.4–30 m/s, accuracy: ± (0.2 m/s +2% of the measurement value); Testo, Germany).

Fresh tea leaves were harvested from the plantations of ENJOY MANOR located in Yingde, Guangdong, China, and were immediately transported to the College of Engineering, South China Agricultural University. Next, the damaged and defective fresh tea leaves were sorted and discarded. Before the test, the air temperature in the container was regulated at approximately 15 ± 1 °C by the refrigerating system for 6 h. During the test, the temperature from the perforated clapboard was set at 14–16 °C, and the air velocity of the perforated clapboard was kept at 1.3 m/s. During the 10 h of the test, the temperatures of the fresh tea leaves were recorded every minute, and the velocities in the ventilation duct of the basket were measured 3 times.

### 2.3. Numerical Method

A mesh model with a total size (length × width × height) of 1500 × 920 × 1350 mm was developed by CATIA V5R20 and then imported using the ICEM CFD program. In order to reduce the workload and shorten the calculation time, an unstructured mesh was adopted, and the mesh type was a triangle/quadrilateral. Details on the boundary conditions in the CFD model built in this paper are shown in Figure 6.

The calculation results and calculation time are affected by the mesh size and number [27,30]. A coarse mesh can reduce the computational cost, while a small mesh size can help enhance the computational efficiency. Therefore, the proper mesh size for a model should be investigated, and a mesh independence test was performed with temperature variations on fresh tea leaves under the same ambient air temperature to study the mesh models with different mesh numbers ranging from 1.3 × 10^6^ cells initially to approximately 5.3 × 10^6^ cells. Meanwhile, the mesh density improved in some areas, such as the outlet, the velocity inlet, and the product, to enhance the computational accuracy in the simulation of these components.

Figure 7 shows the product temperature at the 4 evaluation locations. The product temperature slightly fluctuated as the number of meshes increases. The curve of the fresh tea leaves’ temperature trended to be stable when the mesh quantities were in the range of 4.5 × 10^6^ to 5.3 × 10^6^. Therefore, the number of cells for such mesh models ranged from 4.5 × 10^6^ to 5.3 × 10^6^. The minimum equiangle skewness was 0.268, and the minimum mesh mass was 0.317. The largest and smallest grid units were 0.07 and 0.003 m, respectively.

### 2.4. Mathematical Model

In order to reduce the computational and time cost, the mesh model in this study was simplified with several hypotheses [31]:

The products were substituted by a porous medium [32,33]. The effect of ripening or senescence on the respiration of the fresh tea leaves was ignored;

The air in the fresh-keeping container was considered incompressible gas, which ignores the volume change of the gas due to the temperature changes, and this satisfied the hypothesis of Boussinesq at the same time;

The basket in this study was considered as an adiabatic and sealed structure;

The property parameters of the airflow and the fresh tea leaves characteristics were considered as constant.

The following equations were the control equations in the CFD [34]:

(1) Continuity equation:(1)$$\frac{\partial \rho_{f}}{\partial t}+\nabla \cdot (\rho_{f}v)=S_{m}$$
where $\rho_{f}$ is the fluid density in kg m^−3^; *v* is the velocity vector in m s^−1^; *S_m_* is the quality of the source term in kg m^−3^ s^−1^. The increase in mass should be equal to the mass flux. The increment in mass should be equal to the mass flux density. The buoyancy force [35] increases as a result of density variation as per the assumptions that only the effects of the buoyancy term and the temperature on fluid density were considered; other effects were ignored [36].

(2) Momentum equation:(2)$$\frac{\partial (\rho_{f}v)}{\partial t}+\nabla \cdot (\rho_{f}vv)=-\nabla p+\rho_{f}g+F$$
where *p* is the static pressure in Pa; $\rho_{f}g$ is the gravitational term in N m^−3^; *F* is the external force in N m^−3^. The product affects the direction and velocity value of the airflow in a container. In particular, the air velocity can be significantly weakened upon contact with the product [37]. Consideration must be given to the viscosity of the airflow near the container and the product surface size; thus, superficial velocity should be an option in the porous formulation. *S_j_* is the source term for the momentum equation, which is given by:(3)$$S_{j}=\frac{\mu }{\alpha }v_{j}+C_{2}(\frac{1}{2}\rho vv_{j})$$
where $\frac{1}{\alpha }$ is the viscous drag coefficient and *C*_2_ is the inertial resistance coefficient. The wall treatment in this paper was enhanced to consider the effect of wall roughness on the airflow by the wall enhancement function which can be incorporated into the models by establishing a viscous model.

(3) Energy equation
(4)$$\frac{\partial }{\partial t}(\varphi \rho_{f}E_{f}+(1-\varphi )\rho_{p}E_{p})+\nabla \cdot (v(\rho_{f}E_{f}+p))=\nabla \cdot k_{eff}\cdot \nabla T-\sum_{i}^{}h_{i}J_{i}+S_{i}^{h}$$
where φ is the porosity of the medium; $E_{f}$ is the fluid energy in J kg^−1^; $E_{p}$ is the energy of tea in J kg^−1^; $\rho_{p}$ is the density of tea in kg m^−3^; $k_{eff}$ is the effective thermal conductivity of the porous term in W m^−1^k^−1^; $h_{i}$ is the enthalpy of species ‘i’ in J kg^−1^; T is temperature in K; $S_{i}^{h}$ is the heat source in W·m^−1^.

The respiratory heat from the products is the internal heat source and can be expressed by Equation (5). Because the moisture content of fresh tea leaves undergoes a small change during the fresh-keeping stage, the loss of water during respiration was ignored.
(5)$$q_{l}=0.01068R_{CO_{2}}$$
where $q_{l}$ is the respiratory heat of products, kj kg^−1^h^−1^; $R_{CO_{2}}$ is the respiratory strength of fresh tea leaves, mg kg^−1^ h^−1^.

### 2.5. Boundary Conditions

As an important boundary parameter of the simulation, the fluid type is determined by the Reynolds number (Re) [37]. The Reynolds number of the model was calculated using Equation (5) [38].
(6)Re=ρvdμ
where Re is the Reynolds number; ρ is the density of the fluid (the density of air is 1.225 kg m−^3^); d is the characteristic length in m; v is the inlet velocity in m s^−1^; μ is the viscosity coefficient in m^2^ s^−1^.

In previous CFD studies on refrigerated food applications [39], the k-ε turbulent model was widely used to solve many turbulence problems with high Re values, because it supplies adequate information regarding the turbulent process [40,41,42]. A standard k-ε model reduces the computational cost in terms of memory and CPU time [43]. The turbulence intensity (I) was needed for the CFD simulation, which was calculated by Equation (6) [37]:(7)I=0.16(ReDH)−18
where ReDH is the Reynolds number calculated by hydraulic diameter.

The 35 small holes on the mesh model of the perforated clapboard were set as the inlet of the model, which was also set as the velocity inlet. The velocity was constant for both inlets according to the boundary condition obtained by Equations (1) and (2), which in the inlet boundary, were set to υy = V, υx = υz = 0. The air duct was set as the outlet of the mesh model, which was also set as the outlet-vent, and the turbulence intensity and hydraulic diameter were set in accordance with the inlet boundary condition.

The basket with fresh tea leaves was regarded as the products area and considered as a porous medium with a porosity of 0.491. The metabolism of fresh tea leaves continues after harvest, and the heat added via product respiration is a non-negligible portion of the heat load. Therefore, in the numerical simulation, the respiration heat of fresh tea leaves needs to be considered, and a heat source term was added to the models. The resistance of cold air passing through the porous medium was obtained by the ventilation resistance device, as shown Figure 5. The specific physical parameters in the model are shown in Table 1.

Before the model calculation, a steady solver was adopted to obtain the initial flow field conditions, which can reduce the calculation time. Next, a transient solver was adopted to calculate such models to obtain the change in the flow field. The initial inlet temperature was set to be the same as the ambient temperature, and the product temperature was set at 28 °C. A method of the SIMPLE algorithm was applied to the pressure–velocity coupling, and a least squares cell-based method was employed in the gradient computation. The pressure, momentum, and energy were solved by second-order discretization schemes, but the turbulent kinetic energy and the turbulent energy dissipation rate were solved by the first-order discrete scheme. This paper focused on the temperature field distribution and core temperature of fresh tea leaves during transportation; therefore, it was solved based on a time-dependent solver. The maximum temperature point and average temperature of the products at the monitoring points were recorded to show the changes in the product temperature over time. The model calculations were conducted by a computer equipped with Windows 10 with 64 bits, an Intel^®^ Core i9-9900 CPU, and 32 Gb of RAM.

### 2.6. Error Calculation

Some of the results of the simulations were verified by comparing the simulated value to the test value. The calculation methods are shown as Equations (8)–(11) including root-mean-square error (*RMSE*), mean relative error (*MRE*), relative error (*RE*), and standard deviation (*SD*). For *Xi* (*i* = [1, *N*]), these calculation errors are defined as:(8)$$RMSE(X)=\sqrt{\frac{1}{N}\sum_{i=1}^{N}{\left(X_{sim,i}-X_{exp,i}\right)}^{2}}$$
(9)$$MRE(X)=\frac{1}{N}\sum_{i=1}^{N}\left|\frac{X_{sim,i}-X_{exp,i}}{X_{exp,i}}\right|100\%$$
(10)$$RE\left(X\right)=\left|\frac{X_{sim,i}-X_{exp,i}}{X_{exp,i}}\right|100\%$$
(11)$$SD(X)=\sqrt{\frac{\sum_{i=1}^{N}{\left(X_{i}-X_{avg}\right)}^{2}}{N}}$$

## 3. Results and Discussion

### 3.1. Deriving the Forecasting Coefficients of the Product’s Resistance

The experimental data from the product resistance test and the fitted equation of the resistance are shown in Figure 8. The coefficient of determination (R^2^) was 95.49%, which is sufficiently precise to present the experimental data. According to the experimental data, the inertial resistance ($\frac{1}{\alpha }$) was 103.68, and the viscous resistance (C2) was 2.381 × 10^6^. Data collected were transformed to the form as per Equations (3) and (12).
(12)$$\Delta p=av^{2}+bv$$

### 3.2. Model Accuracy Analysis

A comparison between the simulated and test values of the temperature of fresh tea leaves in the basket located at different positions is shown in Figure 9. The trends in the simulated temperature and test temperature were essentially the same. The calculation errors between the simulated and test temperature values were obtained by simultaneously using Equations (8) and (9) and are presented in Table 2. The results indicate that the maximum RMSE was 1.158 °C and the maximum MRE was 5.410%. Thus, the test and simulation values were in good agreement. These confirmed the reliability and accuracy of the numerical predictions. The simulated temperature was generally larger than the test temperature; this may be caused by the deviation between the accuracy of the sensor and the sensor distribution. In addition, the variations in the respiration heat from the fresh tea leaves, which is related to temperature variations, were not considered in this paper [30,46].

A comparison between the simulated value and the test value of velocity at each monitor point is shown in Figure 10. The relative calculation errors between the simulated and test velocity values were obtained by the simultaneous use of Equations (10) and (11) and are presented in Table 3. The results indicate that the maximum deviation between the simulated velocity and test velocity was 0.11 m/s, the maximum RE was 29.05%, and the maximum SD was 0.024. Such a high value for the RE at certain locations may be attributed to the poor accuracy of measurements at low velocity, where a small difference induces a high relatively error [32]. Moreover, one restriction of the sensor is that it can only measure the velocity in one direction without considering the anisotropy of airflow.

### 3.3. Effects of Different Ambient Temperature

The ventilation duct, vent hole areas, and the bulk density were set as square 100 × 100 mm, 7.5%, and 150 kg/m^3^, respectively. The effects of different ambient temperatures (i.e., 10, 20, and 30 °C) on the temperature variations of the fresh tea leaves were investigated, and the results are shown in Figure 11.

The average temperature of the product under different ambient temperatures is shown in Figure 11a. The average temperature of the product was always higher than the ambient temperature due to the respiration heat accumulation from the fresh tea leaves. The decreasing rate of the temperature of the fresh tea leaves increased with the increase in the temperature difference between the tea’s initial temperature and the ambient temperature. The maximum temperature of the product under different ambient temperatures is shown in Figure 11b. A brief rise in the maximum temperature of the product may have been caused by the low internal temperature gradient of the product and the respiration heat accumulation of the tea at the beginning of the transportation process [20]. With the gradual decrease in the surface temperature of the product, the internal temperature gradient increased, which led to the increasing heat transfer. In addition, the internal temperature gradient of the product increased with a lower ambient temperature, which can accelerate the cooling rate of fresh tea leaves. Similar results are seen by Wang et al. (2021).

### 3.4. Effects of Different Ventilation Duct

The ambient temperatures, vent hole areas, and the bulk density were set to 20 °C, 7.5%, and 150 kg/m^3^, respectively. The effects of different ventilation ducts (i.e., no ventilation duct; square 60 × 60 mm; square 80 × 80 mm; square 100 × 100 mm; and round Φ100 mm) on the temperature variations of the fresh tea leaves were investigated, and the results are shown in Figure 12.

It can be seen from Figure 12 that the plastic basket with the ventilation duct could efficiently decrease the temperature of the fresh tea leaves. According to Zhang et al. [47], the temperature variation of fresh tea leaves is related to the heat generation rate, the ventilation volume, and the heat transfer area. The heat exchange rate is improved by increasing the size of the ventilation duct, which enhances the heat transfer area between the air and product. The total heat transferred increases due to the heat transfer area’s enlargement [48]. Furthermore, the cooling effect of fresh tea leaves improves with the increase in the size of the ventilation duct [49]. The cooling effect on fresh tea leaves in the basket with a circular ventilation duct (Φ100 mm) was poorer than those in the basket with a square ventilation duct (100 × 100 mm) but close to those with an 80 × 80 mm ventilation duct, which implies that the heat transfer area between the product and ambient air has a more significant effect than the shape of the ventilation duct on the cooling effect.

### 3.5. Effects of Different Vent Hole Areas

The ambient temperatures, ventilation duct and the bulk density were set to 20 °C, square 100 × 100 mm, and 150 kg/m^3^, respectively. The effects of different vent hole areas (i.e., 0%, 7.5%, and 15%) on the temperature variations of fresh tea leaves were investigated, and the result is shown in Figure 13.

It can be seen from Figure 13 that the three groups of different vent hole areas had little effect on the temperature of the fresh tea leaves. The temperature variation trends of the three groups were almost the same, and the temperature of the fresh tea leaves in the plastic basket with a vent hole was slightly lower than without a vent hole. This may be because the increased heat exchange area due to the vent hole was extremely small compared to the whole surface area of the product, and the air flow exchange improved by the vent holes was greatly affected by the accumulation of fresh tea leaves. Consequently, the effects of the vent hole in that hole area, which was within 15% of the heat exchange between the fresh tea leaves and air, were not significant.

### 3.6. Effects of Bulk Density

The ambient temperatures, ventilation duct, and vent hole areas were set to 20 °C, square 100 × 100 mm, and 7.5%, respectively. The bulk density can indirectly reflect the porosity of the product and has a significant effect on the resistance of air passing through the product [50]. The effects of different bulk densities (i.e., 150, 175, and 200 kg/m^3^) on the temperature variations of the fresh tea leaves were investigated, and the results are shown in Figure 14.

The decreasing rate of the temperature reduced faster, and the temperature of the fresh tea leaves can be seen at a lower bulk density. The reason for this phenomenon may be because the low bulk density product had larger sized free air voids within the fresh tea leaves [51], which promoted the circulation of internal air. The heat transfer area can be improved over the same duration, which enhances the heat transfer between the fresh tea leaves and air. The increase in bulk density had a significant effect on the maximum temperature, while the effect on the average temperature was not significant. With an increase in bulk density, the duration of high temperature is extended, which may lead to the reddening of fresh tea leaves. Smaller air voids might provide higher barriers for air flow compared to bigger air voids, leading to an overall higher resistance during ventilation [52]. Therefore, the bulk density should not be too high. In summary, increasing the bulk density can store more fresh tea leaves, but it delays the temperature decrease in fresh tea leaves and increases the heat in the product via respiratory heat accumulation from the fresh tea leaves.

### 3.7. Effects of the Stack Plastic Basket

The ambient temperatures, bulk density and vent hole areas were set to 20 °C, 150 kg/m^3^, and 7.5%, while the plastic baskets were stacked from a single layer to three layers and were set as non-ventilation ducts or square 100 × 100 mm ventilation ducts. A single layer of a plastic basket was also set as square 100 × 100 mm ventilation ducts and non-ventilation ducts.

It can be seen from Figure 15 that increasing the number of layers in the plastic basket had a significant effect on the temperature variation of the fresh tea leaves. The effect of the stack may be as significant as the basket design (such as adding a ventilation duct and the ventilation duct size) and may influence each other [53]. When the plastic baskets were stacked, the temperature of the fresh tea leaves had a smaller decreasing rate. Meanwhile, the temperature difference of the fresh tea leaves in the baskets with or without the ventilation duct was improved; the maximum temperature difference was 0.52 and 0.40 according to the stacked or single-layer products, respectively. The plastic basket with a ventilation duct had a more significant effect on heat transfer when the plastic baskets were stacked. The area of the ventilation holes cannot be ignored due to the stacking of the product, which effectively improved the air flow of in the internal product and increased the heat transfer between the air and fresh tea leaves.

The cooling effect was more obvious on the baskets with the added ventilation ducts when the product was stacked. Figure 16a shows that the high-temperature area of the plastic basket with ventilation ducts was smaller and the temperature of the product was lower. The maximum temperature of the product with a ventilation duct was 21.2 °C, and the maximum temperature of the product without a ventilation duct was 21.5 °C. There was good correlation between the temperature distribution and the airflow velocity [22]. A comparison of the airflow of the baskets with or without a ventilation duct is shown in Figure 16b. The airflow velocity of the internal product was small. The addition of the ventilation duct allowed vertical airflow to pass internally through the product and took away some of the heat, which accelerated the heat exchange between the fresh tea leaves and air and effectively avoided the accumulation of respiration heat in the center of product. The rate and uniformity of cooling is affected by the vertical airflow [54,55]. However, the lateral airflow is affected by the resistance of the product, which confirms that the vent hole on the ventilation duct has a small effect on the temperature. The flow rate gradually increased from the bottom to the top of the product—which is due to the fact that the direction of airflow near the bottom region was mainly horizontal and the top region was closer to the air outlet. The flow rate was also affected by the temperature difference between the top and bottom.

## 4. Conclusions

This paper presented a new type of transport basket; its heat characteristics were evaluated using a CFD approach, and the accuracy of the model was verified by experiments. The result of the CFD showed good agreement with the experiments.

The basket with the ventilation ducts obtained a good cooling and ventilation effect. A more obvious effect of heat transfer with the new plastic basket could be seen when the product was stacked. The effect of heat transfer was affected by the ambient temperature, the size of the ventilation duct, the bulk density of the fresh tea leaves, and the stacking of the products. The results shown in this paper provide a better understanding of the relationship between ventilation ducts and transport baskets. A basket with ventilation ducts can be applied in the transportation of various fruits and vegetables in the future.

## Figures and Tables

**Figure 1 foods-11-02178-f001:**
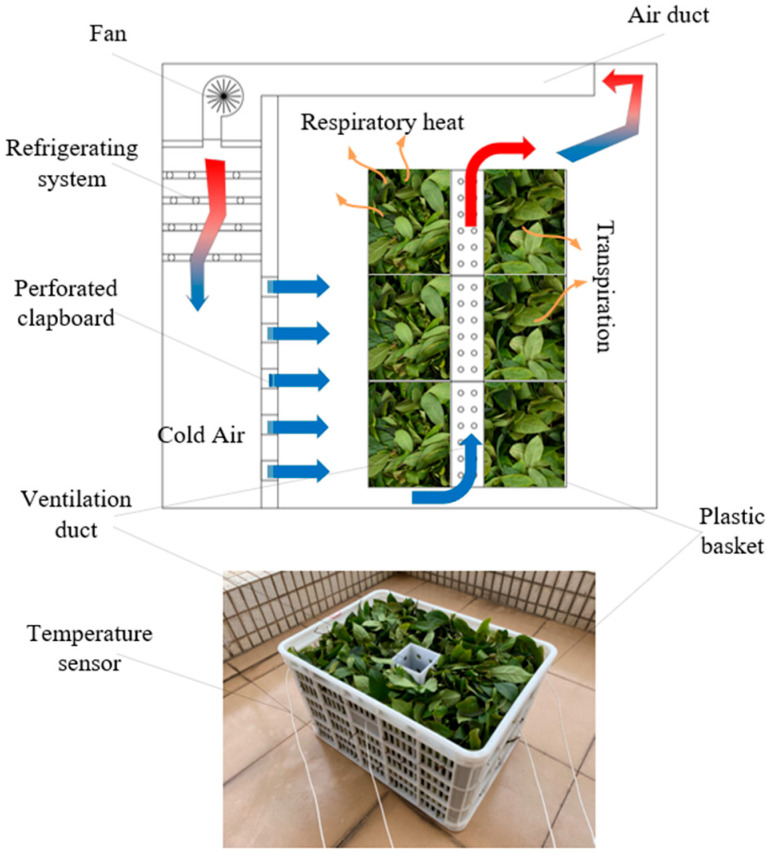
Structure of the fresh-keeping test platform and ventilation basket.

**Figure 2 foods-11-02178-f002:**
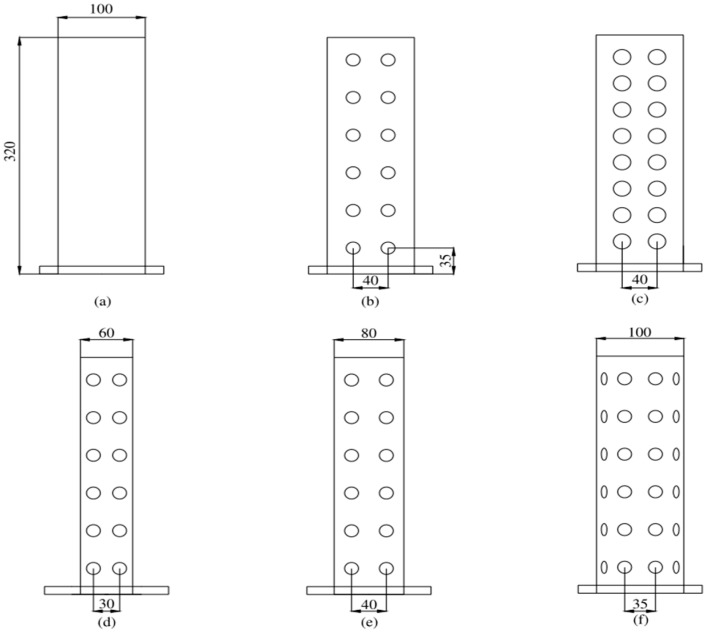
Geometry and diagram of the ventilation duct: (**a**) 0% perforation; (**b**) 7.5% perforation and square 100 × 100 mm; (**c**) 15% perforation; (**d**) square 60 × 60 mm; (**e**) square 80 × 80 mm; and (**f**) round Φ100 mm.

**Figure 3 foods-11-02178-f003:**
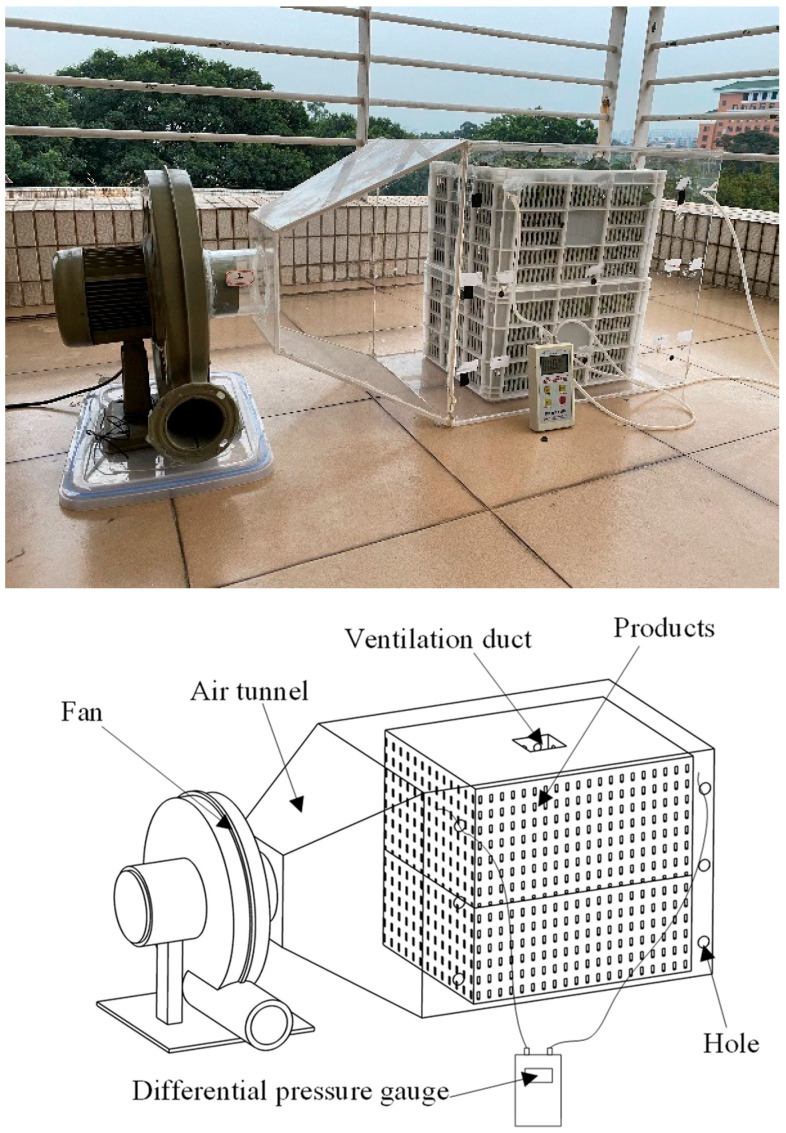
Device used to measure the ventilation resistance.

**Figure 4 foods-11-02178-f004:**
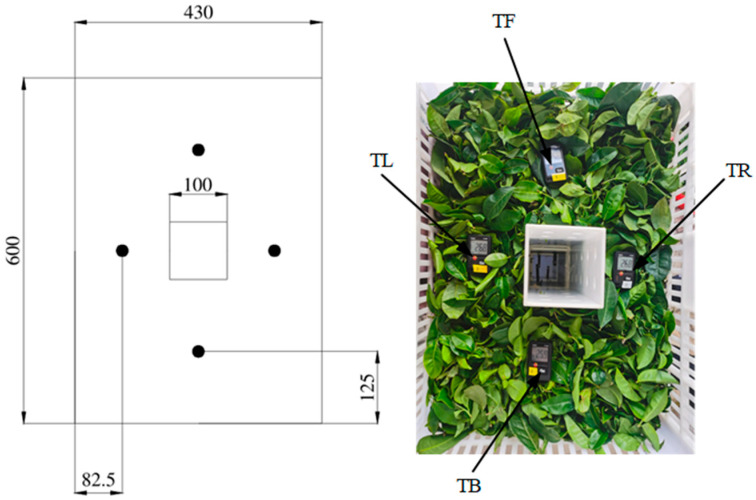
The arrangement of temperature sensors in the basket.

**Figure 5 foods-11-02178-f005:**
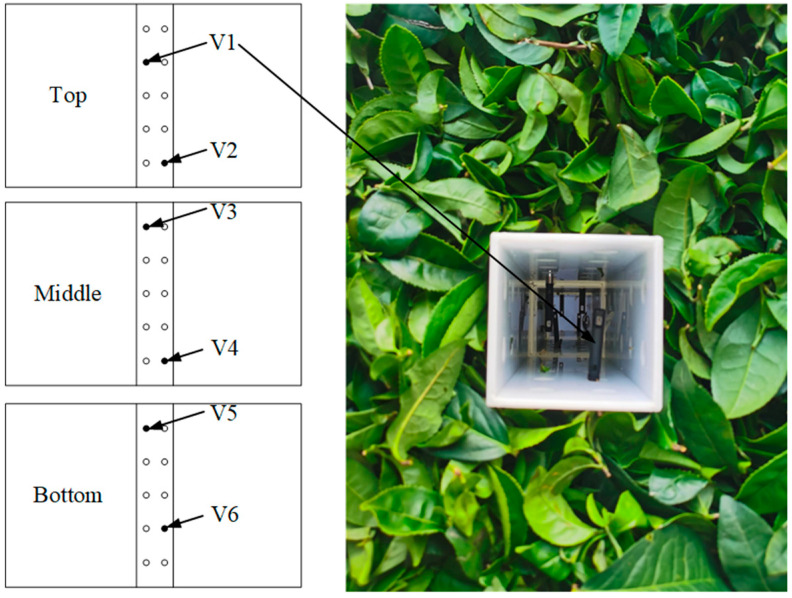
The arrangement of velocity sensors in the ventilation duct.

**Figure 6 foods-11-02178-f006:**
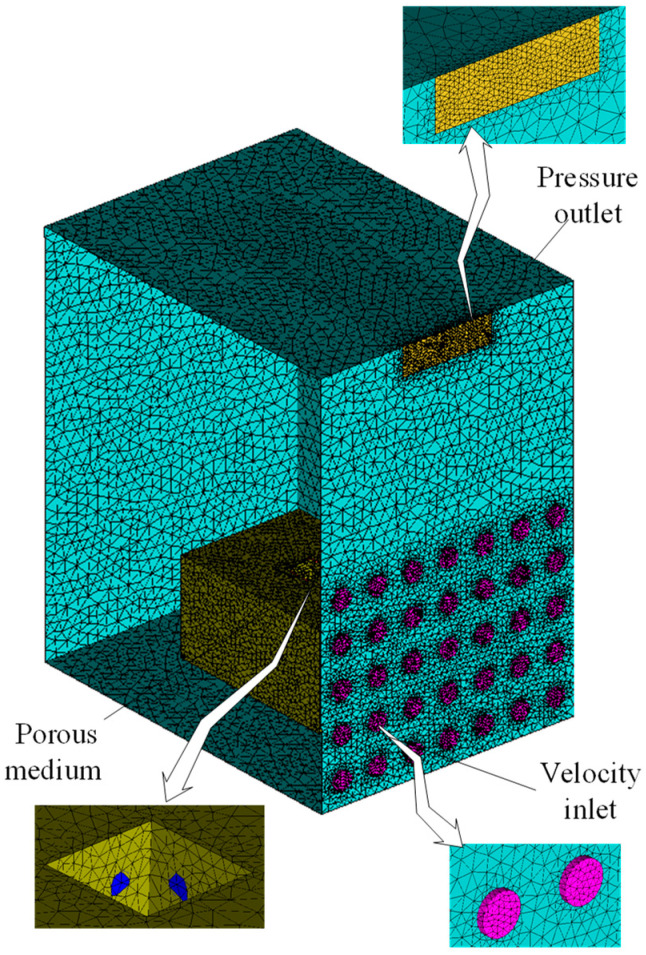
Mesh model.

**Figure 7 foods-11-02178-f007:**
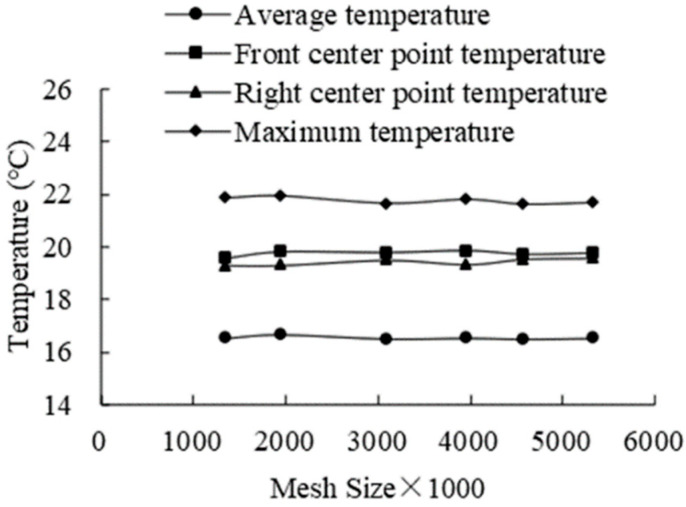
Results of the mesh independence study.

**Figure 8 foods-11-02178-f008:**
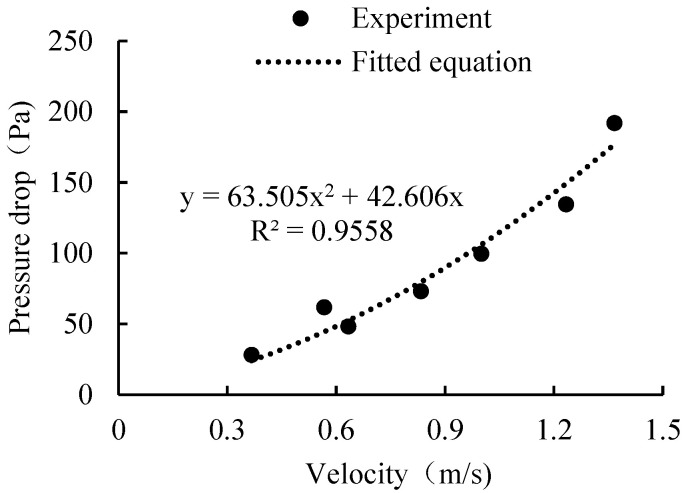
Fitted equation of the porous parameters.

**Figure 9 foods-11-02178-f009:**
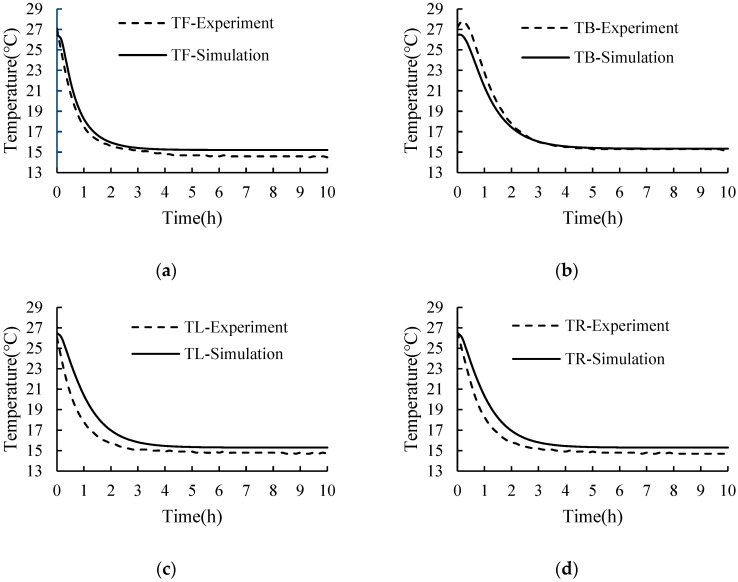
Comparison between the simulation values and the experiment values of the temperatures of fresh tea leaves in the basket: (**a**) front monitor point (TF); (**b**) back monitor point (TB); (**c**) left monitor point (TL); and (**d**) right monitor point (TR).

**Figure 10 foods-11-02178-f010:**
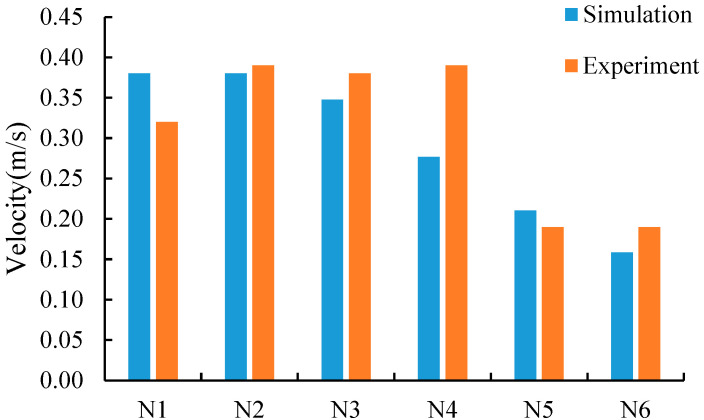
Comparison between the simulation values and experiment values of the air velocity in the ventilation duct.

**Figure 11 foods-11-02178-f011:**
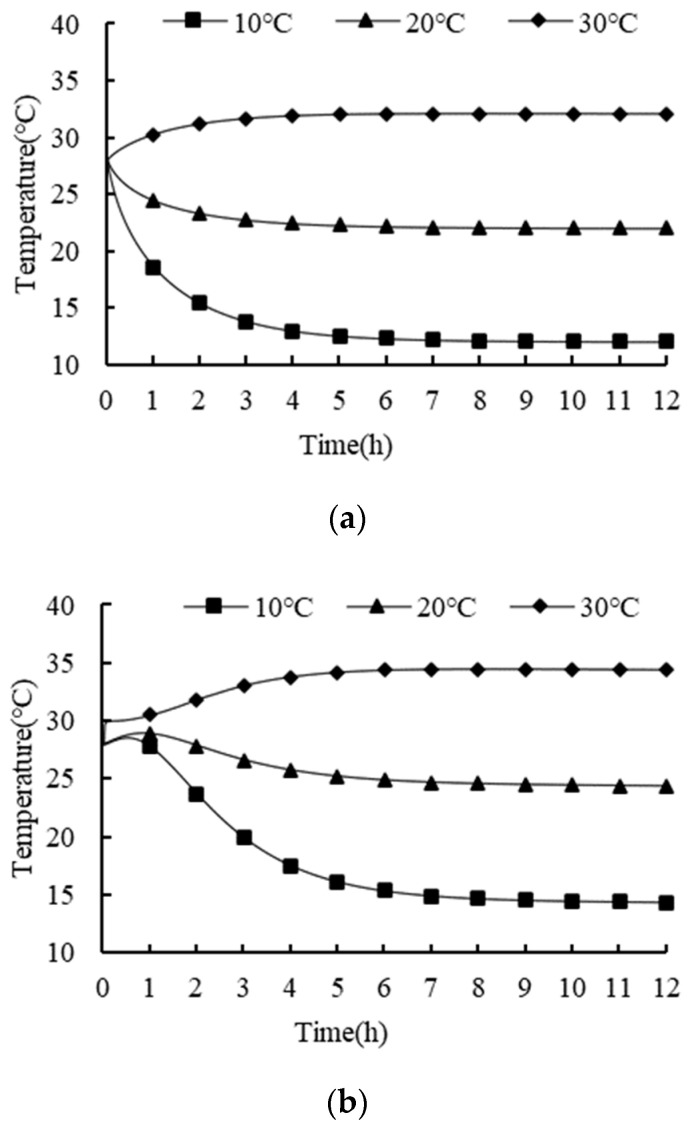
Temperature variations of fresh tea leaves with different ambient temperatures: (**a**) average temperature; and (**b**) maximum temperature.

**Figure 12 foods-11-02178-f012:**
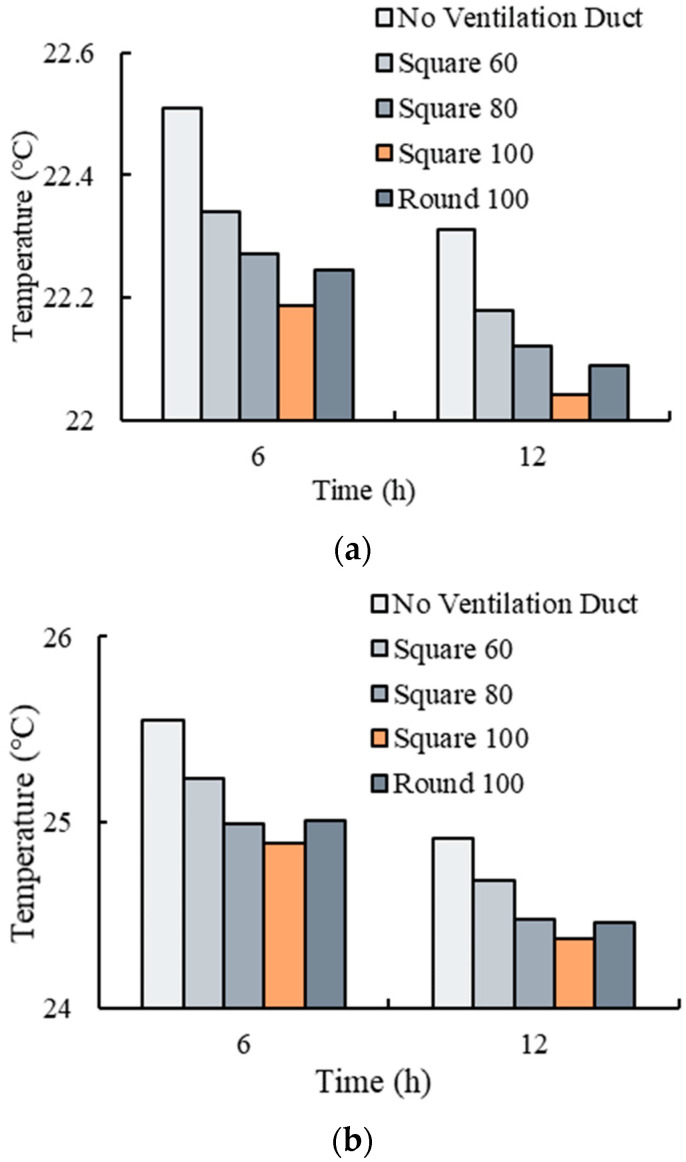
Temperature variations of the fresh tea leaves with different ventilation ducts: (**a**) average temperature; and (**b**) maximum temperature.

**Figure 13 foods-11-02178-f013:**
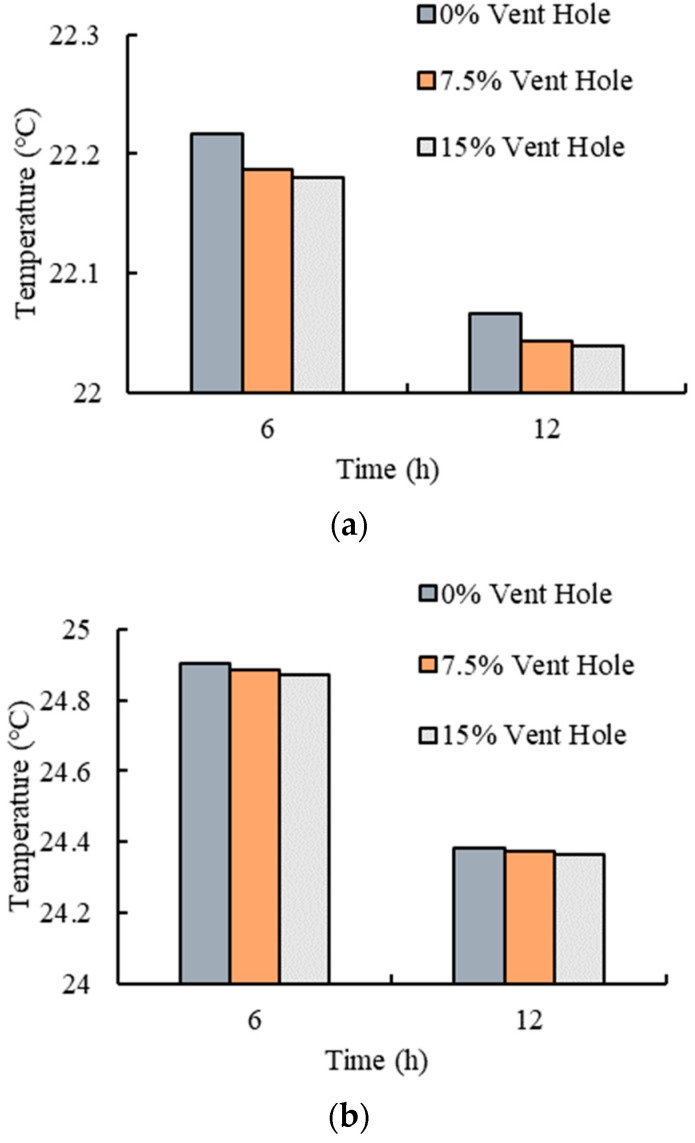
Temperature variations of the fresh tea leaves with different vent hole areas: (**a**) average temperature; and (**b**) maximum temperature.

**Figure 14 foods-11-02178-f014:**
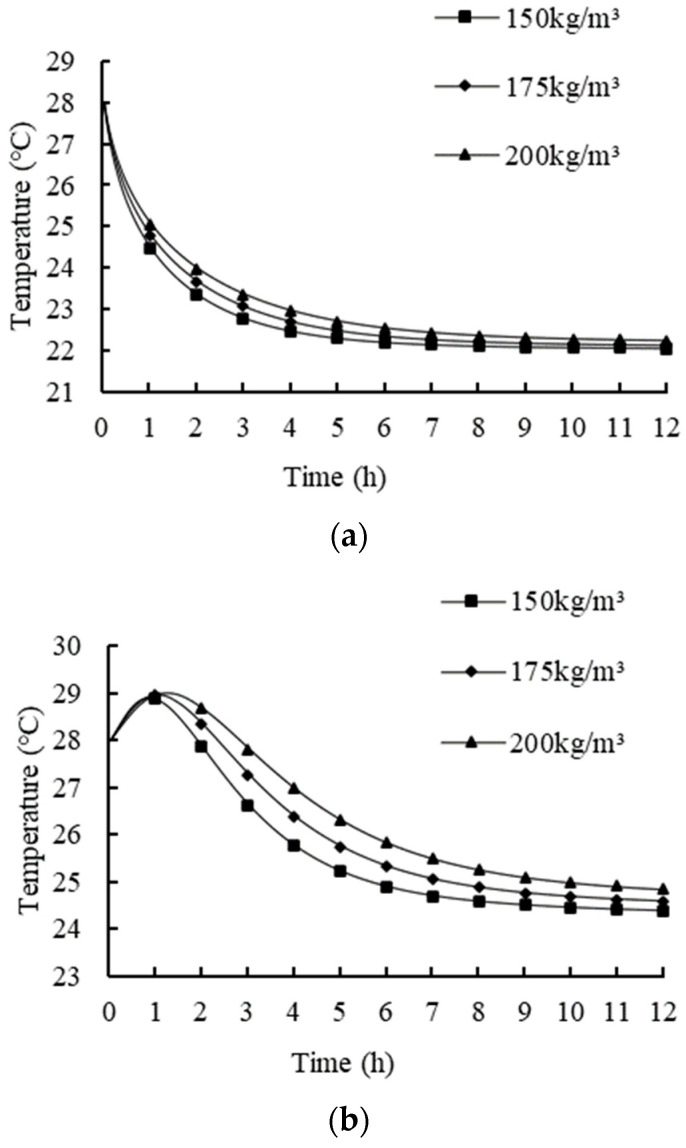
Temperature variations of the fresh tea leaves with different bulk densities: (**a**) average temperature; and (**b**) maximum temperature.

**Figure 15 foods-11-02178-f015:**
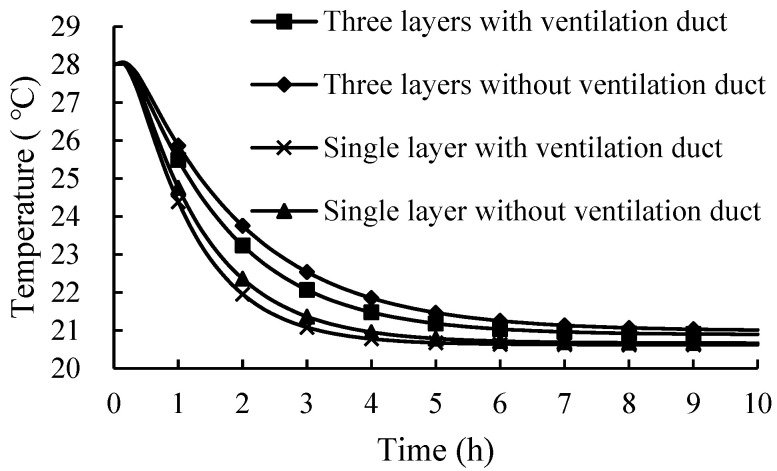
Temperature variations of the fresh tea leaves with different stacks.

**Figure 16 foods-11-02178-f016:**
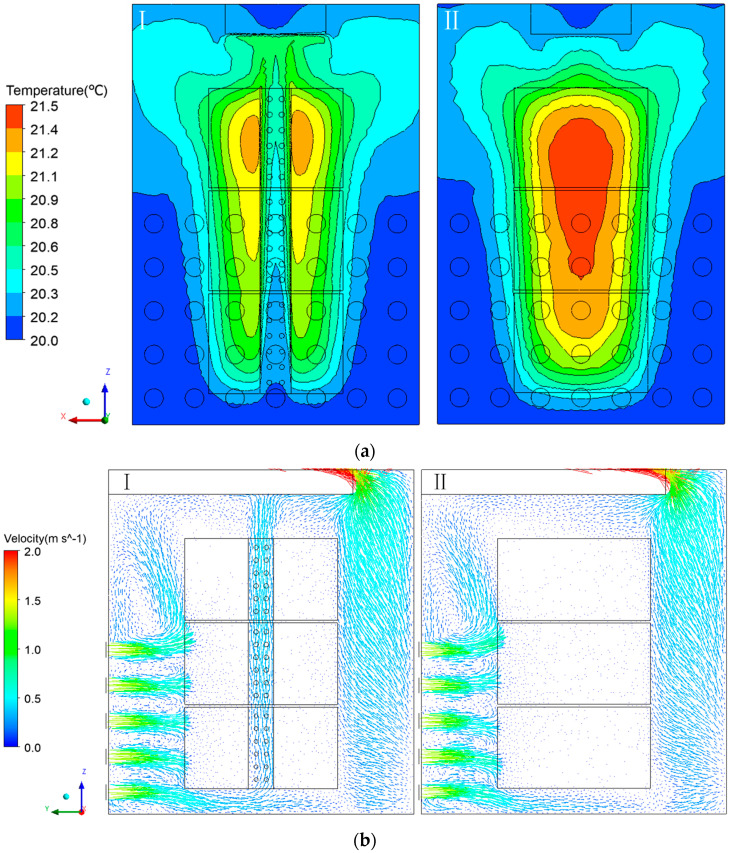
The distribution of the stacking plastic baskets with a ventilation duct and non-ventilation duct (at 10 h of cooling): (**a**) Temperature distribution of the fresh tea leaves (I. plastic basket with a ventilation duct; and II. plastic basket without a ventilation duct). (**b**) Airflow distribution of the fresh tea leaves (I. plastic basket with a ventilation duct; II. plastic basket without a ventilation duct).

**Table 1 foods-11-02178-t001:** Setting of physical parameters.

Name	Parameter	Numerical Value
Air	density/(kg m^−3^) specific heat/(J kg^−1^ K^−1^) heat conductivity coefficient/(W m^−1^ K^−1^) dynamic coefficient of viscosity	1.225 1006 0.0225 1.79 × 10^−5^
Container material [44]	density/(kg m^−3^) specific heat/(J kg^−1^ K^−1^) heat conductivity coefficient/(W m^−1^ K^−1^)	1612 2004.44 0.35
Fresh tea leaves	density/(kg m^−3^) specific heat/(J kg^−1^ K^−1^) heat conductivity coefficient/(W m^−1^ K^−1^) respiratory heat/(W m^−3^)	385 3495 0.493 329.2
Plastic [45]	density/(kg m^−3^) specific heat/(J kg^−1^ K^−1^) heat conductivity coefficient/(W m^−1^ K^−1^)	220 1700 0.048

**Table 2 foods-11-02178-t002:** Comparison between the simulation values and the experiment value of the temperature of fresh tea leaves in the basket.

Monitor Point	RMSE	MRE (%)
TF	0.613	3.639
TB	0.574	1.362
TL	1.158	4.826
TR	0.926	5.410

**Table 3 foods-11-02178-t003:** Comparison between the simulation values and experiment values of the air velocity in the ventilation duct.

The Number of Sensors	V1	V2	V3	V4	V5	V6
Simulation (m/s)	0.38	0.38	0.35	0.28	0.21	0.16
Experiment (m/s)	0.32	0.39	0.38	0.39	0.19	0.19
Relative error (%)	18.75	2.56	8.59	29.05	10.85	16.58

## Data Availability

Data presented in this study are available in the article.

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
