# Peer review of "Numerical Analysis on Heat Characteristics of the Ventilation Basket for Fresh Tea Leaves"

_foods, 2022, doi:10.3390/foods11152178_

Round 1
Reviewer 1 Report
Numerical analysis on heat characteristics of the ventilation
basket for fresh tea leaves.
Reviewer Comments to Authors:
Title: No change needed.
Abstract: Authors should present the numerical values of the results attained from the study. In this form abstract looks like a summary and lacks technicality. Please be specific with the findings for each model and provide the data in the abstract.
Introduction: All the changes have been highlighted and commented in attached reviewer reprt PDF attached.
Materials and Methods: Authors should format the materials and methods by taking a look on the recently published papers of Foods MDPI. Although the contentent and drawings of the equipment used are highly appreciated.
Results and discussion: This section must be improved and standard deviation in all the tables and figures are missing. Please provide the respective number of replication. Pattern of graphs and figures is consistant and presennted in best form.
Conclusions: Authors should include the conclusive results to indicate the bebefits of the study for commercialization.
General comments: I recommend minor revision. The paper has novelty for commercialization. However, the english editing service and spell check is highly recomended for upgrading the quality of manuscript. Overall this is a great work.

Reviewer 2 Report
The manuscript deals with numerical analysis on heat characteristics of the ventilation basket for fresh tea leaves.
The English language must be revised.
Please separate values from units, e.g. “1.225 Kg” not “1.225Kg”.
Please replace “Kg” by “kg”.
Please number the lines consecutively.
Abstract
This section is vague. Please present your main results.
Materials and methods
Page 4- “The thermophysical property parameters of airflow and fresh tea leaves are considered as constant.”??Fresh leaves characteristics??
Results and discussion
This section must be improved. The results must be better compared with other studies available in the literature.
Page 10- “The experimental data of product resistance test and the fitted equation of the resistance is shown as Fig.8. The coefficient of determination R2 is 95.49%, which is precise enough to present experimental data.”??the model goodness of fit was only assessed by the coefficient of determination??
Conclusion
Please do not repeat your results and focus on your main conclusions.
Author Response
Response to Reviewer 2 Comments
Dear Editors,
Thank you for your letter and for the Reviewers’ comments concerning our manuscript entitled “Numerical analysis on heat characteristics of the ventilation basket for fresh tea leaves”. Those comments are all valuable and very for revising and improving our paper, as well as the important guiding significance to our researches. We have studied comments carefully and had made correction which we meet with approval. We tried our best to improve the manuscript and made some changes in the manuscript. These changes will not influence the content and framework of the paper.
Kind regards,
Dr. Guo
The main corrections in the paper and the responds to the Reviewer’ comments are as follow:
Point 1: The English language must be revised.
Response 1: The manuscript has completed language editing to upgrading the quality.
Point 2: Please separate values from units, e.g. “1.225 Kg” not “1.225Kg”..
Response 2: Values and units have been separated in the manuscript.
Point 3: Please replace “Kg” by “kg”.
Response 3: The “Kg” was changed to “kg”.
Point 4: Please number the lines consecutively.
Response 4: The issue has been addressed in the study.
Point 5: This section is vague. Please present your main results.
Response 5: The abstract was revised according to the references. The numerical values of the results attained from the study was added in the abstract.
Point 6: I recommend minor revision. The paper has novelty for commercialization. However, the english editing service and spell check is highly recomended for upgrading the quality of manuscript. Overall this is a great work.
Response 6: The manuscript has completed language editing to upgrading the quality.
Point 7: Page 4- “The thermophysical property parameters of airflow and fresh tea leaves are considered as constant.”??Fresh leaves characteristics??
Response 7: The sentence was revised as “The property parameters of airflow and fresh tea leaves characteristics are considered as constant”, which was marked in red in page 8. Fresh leaves characteristics was shown as Table 1.
Point 8: This section must be improved. The results must be better compared with other studies available in the literature.
Response 8:
The standard deviations of the experiment value were added in the table 3 and figure 10, which was marked in red in page 13. The replications of experiment was indicated page 6 and marked in red.
A reference was added in this manuscript in page 14 and marked in red.
A reference was added in this manuscript in page 15 and marked in red.
Two references were added in this manuscript in page 19 and marked in red.
A paragraph in this section was revised and marked in page 19 in red.
Point 9: Page 10- “The experimental data of product resistance test and the fitted equation of the resistance is shown as Fig.8. The coefficient of determination R2 is 95.49%, which is precise enough to present experimental data.”??the model goodness of fit was only assessed by the coefficient of determination??
Response 9: We have read a lot of references to understand the measurement and evaluation methods of airflow resistance. These scholars assessed the fitting of the model by the coefficient of determination. References include “Characteristic analysis of humidity control in a fresh-keeping container using CFD model” by Guo, “Analysis of airflow and heat transfer inside fruit packed refrigerated shipping container: Part II – Evaluation of apple packaging design and vertical flow resistance” by Getahun and “Resistance to airflow and cooling patterns through multi-scale packaging of table grapes” by Ngcobo.
Point 10: Please do not repeat your results and focus on your main conclusions.
Response 10: The conclusion section was revised to focus on the main conclusions, which was marked in page 19-20 in red.
For a new version of the manuscript, please see the attachment
We appreciate for Editors warm work earnestly, and hope that the corrections will meet with approval. If you have any question about this paper, please don’t hesitate to contact us.
Once again, thank you very much for your comments and suggestions.

Round 2
Reviewer 2 Report
The manuscript was improved.